# Development of a Novel UPLC-MS/MS Method for the Simultaneous Determination of 16 Mycotoxins in Different Tea Categories

**DOI:** 10.3390/toxins14030169

**Published:** 2022-02-24

**Authors:** Haiyan Zhou, Zheng Yan, Song Yu, Aibo Wu, Na Liu

**Affiliations:** 1SIBS-UGENT-SJTU Joint Laboratory of Mycotoxin Research, CAS Key Laboratory of Nutrition, Metabolism and Food Safety, Shanghai Institute of Nutrition and Health, University of Chinese Academy of Sciences, Chinese Academy of Sciences, Shanghai 200030, China; zhouhaiyan2018@sibs.ac.cn (H.Z.); zyan@sibs.ac.cn (Z.Y.); abwu@sibs.ac.cn (A.W.); 2Division of Chemical Toxicity and Safety Assessment, Shanghai Municipal Center for Disease Control and Prevention, Shanghai 200336, China; yusohar@163.com

**Keywords:** mycotoxins, tea, ultra-performance liquid chromatography-tandem mass spectrometry, simultaneous determination, purification

## Abstract

The contamination of potential mycotoxins in tea production and consumption has always been a concern. However, the risk monitoring on multiple mycotoxins remains a challenge by existing methods due to the high cost and complex operation in tea matrices. This research has developed a simple ultra-performance liquid chromatography-tandem mass spectrometry strategy based on our homemade purification column, which can be applied in the detections of mycotoxins in complex tea matrices with high-effectively purifying and removing pigment capacity for 16 mycotoxins. The limits of detection and the limits of quantification were in the ranges of 0.015~15.00 and 0.03~30.00 µg·kg^−1^ for 16 mycotoxins, respectively. Recoveries from mycotoxin-fortified tea samples (0.13~1200 µg·kg^−1^) in different tea matrices ranged from 61.27 to 118.46%, with their relative standard deviations below 20%. Moreover, this method has been successfully applied to the analysis and investigation of the levels of 16 mycotoxins in major categories of tea and the monitoring of multiple mycotoxins in processed samples of ripened Pu-erh. In conclusion, the proposed strategy is simple, effective, time-saving, and low-cost for the determination of a large number of tea samples.

## 1. Introduction

Tea, the leaves of *Camellia sinensis*, is divided into green, white, yellow, oolong, black, and dark tea, depending on the type of fermentation process [1], which results in the differences of health-promoting components [2]. In a word, tea has attracted increasing attention for its unique human health benefits [3,4,5].

Tea products take a long time to process, with stages including planting, primary and refined processing, packaging, storage, and transportation for final consumption. Mycotoxins, potential hazardous pollutants generally produced by toxigenic fungi, are of great concern during manufacturing [6,7,8,9,10]. Furthermore, it is known that post-fermented tea contains many microorganisms. *Aspergillus* spp., *Fusarium* spp. and *Penicillium* spp. are the most prevalent microbial groups present during the processing and storage of post-fermented tea [11,12]. Some fungi are important contributors to tea manufacture, promoting the formation of active ingredients and the unique aroma of tea during the pile fermentation period. However, there are concerns that some fungal species might be mycotoxin producers and cause mycotoxin accumulation under suitable environmental conditions [13,14]. Currently, mycotoxins in foodstuffs are receiving increasing attention for their toxicity, carcinogenicity, teratogenicity, mutagenicity, genotoxicity, and immunotoxicity to humans and other animals. Some studies have focused on the occurrence of mycotoxin pollutants in tea, containing zearalenone (ZEN), aflatoxins (AFs), and ochratoxin A (OTA) in dark tea [14,15] and AFs, deoxynivalenol (DON) in black, green, and other teas [16,17]. However, there are no international maximum regulation limits (MRLs) for mycotoxins in tea due to the limited contamination data and exposure assessments in different types of tea, especially for the new toxicities of common mycotoxins or emerging mycotoxins.

Concerning the presence of multiple mycotoxins in a single tea, several studies have focused on the quantitative determination of multiple mycotoxins in a few kinds of tea through high-performance liquid chromatography (HPLC) and liquid chromatography-tandem mass spectrometry (LC-MS/MS) [14,17,18,19]. Tea samples, complex matrices containing abundant caffeine and polyphenols, are considered to be analytically challenging in trace pollutant residue detection methods [20,21]. The sample pretreatment processes of existing quantitative methods for mycotoxins in complex tea matrices are complicated and expensive, such as dispersive liquid–liquid microextraction (DLLME), which is based on isotopes [18]; solid phase extraction (SPE), which is based on salting out; dispersive solid-phase extraction (D-SPE) during extraction [22]; and the successive use of NH_2_-SPE and C_18_-SPE [19], multiple immunoaffinity columns (IACs) [17], and MFCs (multifunctional columns) integrated with IACs [14] in the clean-up stage. It is still a challenge to establish simple pretreatment processes to eliminate matrix effects (MEs) with broad applicability to monitor the potential multiple mycotoxins in various tea matrices.

The present study aimed to develop and validate an ultra-performance LC-MS/MS method with a homemade purification column to determine 16 mycotoxins in four categories of tea (green, oolong, black, and dark tea). Additionally, the occurrence and contamination of multi-mycotoxin in various tea samples were investigated with the newly established and successfully validated [23,24] method.

## 2. Results and Discussion

### 2.1. UPLC-MS/MS Analysis Conditions

The target mycotoxins in this paper are small molecules with a wide polarity range (log Kow = −0.85~5.37). C_18_ reversed-phase liquid chromatography columns are the most commonly used columns for the separation of mycotoxin (Table 1). During the application of the method, [M + Na]^+^ ions can lead to incomplete cleavage of parent ions and reduce the signal abundance of the target analyte [7]. When mobile phase A is ammonium acetate, it can avoid the formation of [M + Na]^+^ in some mycotoxins. In other words, 5 mM ammonium acetate can enhance the response of 15-Ac DON, NEO, and T-2 by promoting the [M + NH_4_]^+^ ion intensities [25]. AFB_1_, AFB_2_, AFG_1_, AFG_2_ and OTA contain methoxy or carbonyl groups that can result in a high abundance of the [M + H]^+^ peak in positive ionization mode. ZEN and its modified forms easily generated [M − H]^−^ ions at *m*/*z* values of 317, 319 and 321. The [M + CH_3_COO]^−^ ions among the precursor ions of DON and 3-Ac DON had the highest ion abundance. Furthermore, the main form of the precursor ions of 15-Ac DON, NEO and T-2 was [M + NH_4_]^+^. The highest CIT ion abundance was in the [M + OH]^−^ ion form. The signal acquisition information of various mycotoxins in the optimized SRM mode, including characteristic ion pairs and collision energy parameters, is shown in Table 2. According to the abundance of ions, we selected appropriate ions as quantitative ions. The total ion chromatography separation effect of the 16 mycotoxins at middle concentrations (Appendix A) under the optimal UPLC-MS conditions is shown in Figure 1.

### 2.2. Improvement of Sample Pretreatment Processes

#### 2.2.1. Optimization of Mycotoxins Extractant in Tea Samples

In our preliminary research, the extraction recoveries of the target analyte in tea gradually increased with increasing acetonitrile concentration. The ME of acetonitrile is the weakest among commonly used extraction agents. However, the extraction effect of mycotoxins sensitive to the polarity range was not ideal when only acetonitrile was used as the extraction agent. Therefore, the extraction recoveries of mycotoxins achieved by acetonitrile with different formic acid contents were considered and compared. The extraction recoveries of OTA, T-2 and NEO were close to 100%, with 5% formic acid in the extractant (Figure 2A). The recoveries of ZEN and its modified forms did not increase gradually with increasing acidity of acetonitrile but were inhibited to less than 60% at an acidity percentage greater than 5%, which may have been due to the influence of hydrogen ions on the response of negative ions. Recoveries of DON and its derivatives were better in acetonitrile with formic acid concentrations from 0 to 2%. The recoveries of AFs were better under extraction with acetonitrile containing formic acid. The extraction recoveries of CIT increased with increasing formic acid concentration, while some recoveries exceeded 120%. When the concentration of formic acid was 1%, the recovery was close to 100% (*p* < 0.05). To monitor the potential risk of multiple mycotoxins in tea simultaneously and improve the wide applicability compared with other established methods, acetonitrile with 1% formic acid was finally selected as the extraction agent to meet the recoveries and responses of the mycotoxins mentioned above (Figure 2A).

#### 2.2.2. Evaluation of Purification Effects on Mycotoxins in Tea Samples

In recent years, multi-functional purification columns have been popular in LC-MS/MS preprocessing for their simple and fast impurity removal. However, the detection cost is expensive for large numbers of samples. We developed a homemade purification column with excellent purification effects for aflatoxins [26]. However, the purification effects for other mycotoxins are still needed to be further evaluated. In this study, the purification effects of homemade were compared with other commercial purification columns on purification for 16 mycotoxins in tea matrices. MWCNTs-COOH:HLB:SG (our homemade purification column) had distinct advantages in 16 mycotoxins purification in tea matrices. MWCNTs-COOH:HLB:SG has the good impurity removal ability among these columns, and it has a wider range of detection for mycotoxins (Figure 2B and Figure 3C). The Cleanert MC purification column had better purification recoveries for 15-Ac DON, AFs, 3-Ac DON, T-2, NEO, and ZENs (61.24~119.58%). The recoveries of AFs, 3-Ac DON, T-2, NEO, and ZEN by the CNW 301 MFC purification column were 63.38~110.53%. The MFC 260 purification column had the best purification recovery for T-2 among the mycotoxins (85.13%). The purification recoveries of 3-Ac DON, DON, AFs, CIT, T-2, NEO, OTA, and ZENs on the Romerlab MycoSpin 400 were 64.95~116.93%. Our homemade purification column especially achieved ideal purification recoveries (74.26~119.98%) in tea matrices for 16 mycotoxins. Furthermore, the MEs of mycotoxins were improved through purification with our homemade purification column (Figure 3). Most current reports apply the successive use of NH_2_-, C_18_-SPE, multiple IACs, or MFCs-IACs to achieve clean-up in some tea matrices [14,17,18,19,22]. With our homemade purification column, the developed method does not require combining with other commercial columns for sample purification in complex tea matrices, which is mainly due to its high-efficiency and broad-selectivity capacity (Figure 2B and Figure 3). Additionally, the cost price ratios of MWCNTs-COOH:HLB:SG (our homemade purification column), Cleanert MC, MFC 301, MFC 260, and MycoSpin 400 were 1:3.61:13.79:11.82:17.00.

### 2.3. Method Validation

The LODs and LOQs of mycotoxins were in the ranges of 0.015~15.00 and 0.03~30.00 µg·kg^−1^ (Appendix A), which are similar to those obtained with the method developed by Pallarés et al. [18]. There are no uniform limits for mycotoxins in tea. Custom Union countries have regulated a limit of AFB1 in raw tea of 5 µg·kg^−1^ [27]. Argentina set limits for AFB1 and AFs in herbal tea materials at 5 and 20 µg·kg^−1^, respectively [28]. In the European Union, the MRL for OTA in both roasted coffee beans and ground roasted coffee is 5.0 µg·kg^−1^; and the MRLs for DON and ZEN in cereals intended for direct human consumption are 750 µg·kg^−1^ and 75 µg·kg^−1^, respectively [24]. The LODs (or LOQs) of this method are far below the MRLs in foodstuffs described above. Recoveries of this method in different tea matrices ranged from 61.27 to 118.46% for mycotoxins. Furthermore, the intra- and inter-day relative standard deviations (RSDs) were below 20% (0.70~19.95%), which indicates that this strategy can quantitatively determine mycotoxins in tea (Table 3). In short, there are satisfactory data from the recoveries of extraction, purification (Figure 2), and the validation of the prosed method (Table 3). A comparison between our proposed method and previously reported methods was completed on mycotoxins quantitative detections (Table 1). Obviously, our method has the advantages of lower cost, wider applicability with satisfactory recovery rates and LOQs, and has the ability to eliminate interferences of more types of tea for multiple mycotoxins monitoring. 

### 2.4. Occurrence of Mycotoxins in Tea Samples

The mycotoxins in the samples of different types of tea were detected to different degrees, with average concentrations ranging from 0.07 to 958.58 µg·kg^−1^ (Table 4). From green, oolong, and black tea to dark tea, the detection types and ranges of most mycotoxins increased, which may have been related to the differences among these teas in processing methods or biochemical components. However, the total positive rates of DON, AFs, α-ZEL, β-ZEL, OTA, and CIT were 6.25% (5/80), 1.25% (1/80), 6.25% (5/80), 10% (8/80), 5% (4/80), and 1.25% (1/80), respectively.

The co-occurrence rate of DON (10%) and its derivatives in tea samples was lower than the result obtained by Reinholds et al., possibly due to study differences in the samples analyzed [16,29]. Kiseleva et al. found that C. sinensis tea samples were the least contaminated among the herbal tea samples analyzed [30]. The average contents of DON in the 20 oolong tea samples in the present study exceeded the MRL of DON in cereals for direct consumption (750 µg·kg^−1^), with five tea samples having contents between 777.51 and 1181.48 µg·kg^−1^. Mannani et al. and Bogdanova et al. observed average content ranges similar to those in the present study: AFB_1_ (LOD~1.04), AFB_2_ (LOD~0.39), AFG_1_ (0.07~0.62), and AFG_2_ (2.97~8.98) [25,31,32]. Importantly, the average contents of AFB_1_ and AFs in dark tea in this study did not exceed the Argentinean limits of 5 and 20 µg·kg^−1^, respectively. The occurrence of ZEN (2.5%) was similar to that detected in the Ye et al. study [14]. In addition, the average content of ZEN did not exceed the relevant MRL (75 µg·kg^−1^). Interestingly, the contents of β-ZEL and α-ZEL in oolong and black tea were higher than those in the other teas. Pakshir et al. employed HPLC and observed similar incidences of OTA in green tea and black tea to those obtained in the present study: although most of the samples were contaminated with OTA, the contents were lower than the MRL (5 µg·kg^−1^) for OTA in coffee [17]. Moreover, in the present study, the average content of OTA was highest in dark tea (666.45 µg·kg^−1^), with three of the small green orange Pu-erh samples (“Xiao Qing Gan” or “Gan Pu”) containing high concentrations of OTA. Li et al. observed that the content of CIT ranged from 7.8 to 206 µg·kg^−1^ among 107 tea samples, and two samples had high CIT contents (>200 µg·kg^−1^) [33,34]. The range of CIT contents in dark tea is similar to that in our study: one ripened Pu-erh tea sample out of the 80 samples had a CIT content above 200 µg·kg^−1^ (203.76 µg·kg^−1^).

The contents of the 16 mycotoxins during the production steps of ripened Pu-erh were also determined. The results illustrated that 15-Ac DON,3-Ac DON, α-ZAL, DON, NEO, β-ZAL, OTA, T-2 and CIT were not detected in the finished Pu-erh. In addition, seven other mycotoxins were detected at (0.10~72.61 µg·kg^−1^), which were lower than the corresponding MRLs listed above. Specifically, 15-Ac DON, DON, T-2, and NEO reached their highest concentrations in the samples from the first repiling step; the contents of OTA in the raw material, samples from the humidifying stage, and samples from the fifth repiling step were higher than those in the samples from other production stages. In addition, the content of CIT was highest in the samples from the third repiling step. Regarding the total amount of AFs and that of ZEN and its modified forms (i.e., α-ZAL, α-ZEL, and β-ZEL) [35], the contents were lowest in the semifinished Pu-erh and increased in the finished tea, but the total content of AFs was lower than 20 µg·kg^−1^ (Appendix A). It is necessary to monitor mycotoxins in ripened Pu-erh during tea manufacturing, especially during later processing or storage. Such monitoring is conducive to eliminating high-risk tea samples as much as possible.

## 3. Conclusions

Risk monitoring on multiple mycotoxins in tea matrices remains a challenge due to the high cost and complex operation. Compared to most commercial and reported columns, our homemade column is effective for more mycotoxins, which was wider in application. Based on a simple and low-cost sample pretreatment process with the help of our homemade purification column, a universal method for simultaneously detecting 16 mycotoxins in tea matrices was successfully established and validated. It is simple and universal, time-saving, and low-cost for the determination of a large number of tea samples. The proposed method was applied to 80 commercial samples and 25 process samples of ripened Pu-erh to analyze mycotoxin levels. Some MRLs were exceeded (for DON, OTA, α-ZEL, β-ZEL, AFG_2_, or AFs) by the detected contents in some samples. Meanwhile, green tea had the lowest mycotoxin contamination among the four types of tea, which may be related to the fermentation process. This contamination investigation laid a good foundation for subsequent risk assessment consistent with drinking tea. The proposed strategy provides a practical and universal solution for the quantitative determination of multiple mycotoxins in complex tea matrices, which would help to monitor the potential risk of mycotoxins in the fields of tea production and consumption.

## 4. Materials and Methods

### 4.1. Chemicals and Reagents

ZEN (Art. No. Z 2125), α-ZEL (Art. No. Z 0166), β-ZEL (Art. No. Z 2000), α-ZAL (Art. No. Z 0292), β-ZAL (Art. No. Z 0417), DON (Art. No. D 0156), 3-Ac DON (Art. No. A 6166), 15-Ac DON (Art. No. A 1556), OTA (Art. No. O 1877), T-2 (Art. No. T 4887), and CIT (Art. No. C 1017) standards were obtained from Sigma-Aldrich (St. Louis, MO, USA). AFs (Part. No. 10000344) and NEO (Part. No. BRM S 92001) analytical standards were obtained from Romer Lab Biopure™. Cleanert MC (Art. No. LC-MYT10-B) was acquired from Agela Technologies (Shanghai, China). MFC260 (Art. No. M2600-25T) was obtained from Pribolab (Singapore). Romerlab MycoSpin 400 (Art. No. COCMY2400) was purchased from Shanghai Pu Yu Science and Trade, Inc. CNW 301 MFC (Art. No. SBEQ-CD301S) and an analytical-grade hydrophilic-lipophilic balance (HLB; Art. No. SBEQ-CA3100) were purchased from Anpel Laboratory Technologies (Shanghai, China). Analytical-grade carboxyl multiwalled carbon nanotubes (MWCNTs-COOH; Art. No. 190710134732) were acquired from Klamar Reagent, Inc. (Shanghai, China). Acetonitrile comes from Merck (Darmstadt, Germany) is HPLC-grade. Milli-Q quality water (Millipore, Billerica, MA, USA) was used throughout the experiments. Ammonium acetate, formic acid (≥95%) and analytical-grade silica gel (SG; Art. No. 236799) were Sigma-Aldrich products (St. Louis, MO, USA).

### 4.2. Preparation of Stock Solutions

AFB_1_ (2.03 µg·mL^−1^), AFB_2_ (0.503 µg·mL^−1^), AFG_1_ (2.02 µg·mL^−1^), AFG_2_ (0.502 µg·mL^−1^), and NEO (100.5 µg·mL^−1^) stock solutions were obtained as certified solutions in acetonitrile. Stock solutions of α-ZAL, β-ZAL, ZEN, α-ZEL, β-ZEL, DON, 15-Ac DON, 3-Ac DON, T-2, CIT (5 mg·mL^−1^), and OTA (1 mg·mL^−1^) were manufactured by diluting commercially available mycotoxins with pure acetonitrile. The mixture was freshly made and stored in the refrigerator (−40 °C) before use. Tea extract from the blank sample was applied to prepare matrix-matched standards.

### 4.3. Samples Collection

Eighty commercial tea samples were randomly collected from Shanghai Difute International Tea Co., Ltd. (Shanghai, China). The 80 samples comprised 20 samples of green tea, 20 samples of oolong tea, 20 samples of black tea, and 20 samples of dark tea. Additionally, 25 samples of tea materials during processing were gathered from Yunnan Fengqing Longrun Co., Ltd. (Lincang, China), between May 2019 and August 2019, including samples of raw materials; samples of tea in the humidifying stage; samples from the first, second, third, fourth, and fifth repiling steps; and samples of piling-up tea, semifinished and finished Pu-erh. The number of samples in each stage of repiling was three, which are taken from the upper, middle, and lower layers. All samples were ground, divided, and stored before use. Each commercial sample was analyzed twice, and the results are expressed as the mean values.

### 4.4. Sample Preparation

A total of 1.0 g sample was placed in the 10 mL homogenizing centrifuge tube. To optimize the extraction process of multiple mycotoxins, we experimented with acetonitrile with different formic acid contents (0%, 1%, 2%, 5%, 10% and 15%). After ultrasound-assisted extraction (30 min), it was centrifuged (3220 g, 10 min). Optimization of the purification process was achieved by comparing the purification effects of different clean-up strategies. Finally, supernatant (1 mL) was selected to add the MWCNTs-COOH: HLB: SG. The centrifuge tube was vortexed (1 min) and centrifuged (13,800 g, 5 min). Finally, the sample was filtered out with the organic filter membrane (0.22 μm) for subsequent analysis on the newly proposed UPLC-MS/MS system. The recoveries of mycotoxins purification and extraction from tea matrices were compared and evaluated. To prepare spiked samples, each mixture was vortexed (30 s) and incubated (25 °C) for solvent evaporation after mycotoxins were added to the ground tea sample. Purification with other multifunctional purification columns was also performed according to the existing literature [15] and relevant product instructions (Appendix A).

### 4.5. UPLC-MS/MS Analysis

Mycotoxins were monitored on an Ultimate 3000 UPLC system (Thermo Fisher Scientific, San Jose, CA, USA) coupled to a TSQ VantageTM triple stage quadrupole mass spectrometer (Thermo Fisher Scientific, San Jose, CA, USA). Analytes were separated on an Agilent Extend C18 column (150 × 3.0 mm, 3.5 µm, Art. No. 763954-302) for gradient elution with water containing 5 mM ammonium acetate and methanol (0.35 mL·min^−1^, 30 °C, 10 μL) through the following elution program: 15% B (initial), 15% B (0–1 min), 15–25% B (1–3 min), 25–50% B (3–6.5 min), 50–100% B (6.5–12 min), 100% B (12–15 min), 100–15% B (15–17 min), and 15% B (17–20 min).

We monitored mycotoxin standards at middle concentrations in full-scan mode (*m*/*z*: 50–1000). Different optimal parent ions were selected, and different collision energies with argon were used to conduct collision-induced dissociation (CID) to obtain the product ion spectrum. The product ions and the collision energy were automatically optimized on the mass spectrometer. The optimized parameters for monitoring selected reactions (SRM), positive (ESI^+^ 3.0 kV) and negative (ESI^−^ 2.5 kV) ionization modes using mass spectrometry are: capillary temperature 300 °C; vaporizer temperature 250 °C; ion sweep and aux gas pressure are 5 psi; and sheath gas pressure 40 psi.

### 4.6. Method Evaluation and Application

According to the documented guidelines [23], the parameters of the UPLC-MS/MS were verified in four tea matrices containing 16 mycotoxins. The ME was evaluated by analyzing the slopes of two sets of mycotoxin standards prepared with organic solvent and the relevant tea extracts. MEs were calculated according to Equation (1) below, where K_a_ (matrices) and K_b_ (organic solvent) are the slopes of curves, respectively.
ME (%) = 100 × (K_a_/K_b_ − 1)(1)

The limits of detection (LODs) and quantification (LOQs) of this method are the lowest detectable concentrations with the signal-to-noise ratio (S/N) values of 3 and 10, respectively. The proposed strategy was validated at three levels in mycotoxin-spiked different types of tea [24]. Recoveries were obtained through Equation (2) below, where C_ai_ is the actual detection concentration for mycotoxin i-spiked samples, C_bi_ is the actual detection concentration for the nonspiked sample, and C_Ai_ is the theoretical concentration of analytes that was added to the sample. The range of added standard concentrations to mycotoxin-free tea with six replicates was 0.13~1200 µg·kg^−1^ (Appendix A). The precision and accuracy of the proposed strategy were checked through intra- and interday analysis. Intraday precision was assessed via the analysis of six replicate mycotoxin-fortified tea samples at three different concentration levels, and interday precision was evaluated consecutively on six different days. The collected commercial and processed tea samples were analyzed for multiple mycotoxins by the established method.
Recovery (%) =100 × (C_ai_ − C_bi_)/C_Ai_(2)

### 4.7. Data Analysis

Use Thermo Xcalibur Qual Browser 4.0 to recognize UPLC-MS raw data. Mycotoxin recoveries for each solvent mixture and purification method were evaluated using two-way ANOVA at a significance level of 0.05; Tukey’s multiple comparisons testing was used to evaluate the significance of difference within each group. GraphPad Prism 5.0 software is used for all statistical analysis and graphic production.

## Figures and Tables

**Figure 1 toxins-14-00169-f001:**
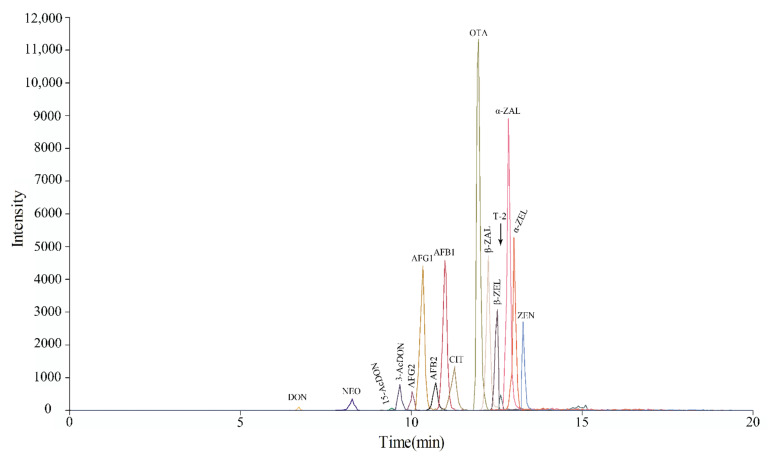
Total ion chromatograms of 16 mycotoxins at middle concentration under optimized chromatographic and mass spectrometry conditions.

**Figure 2 toxins-14-00169-f002:**
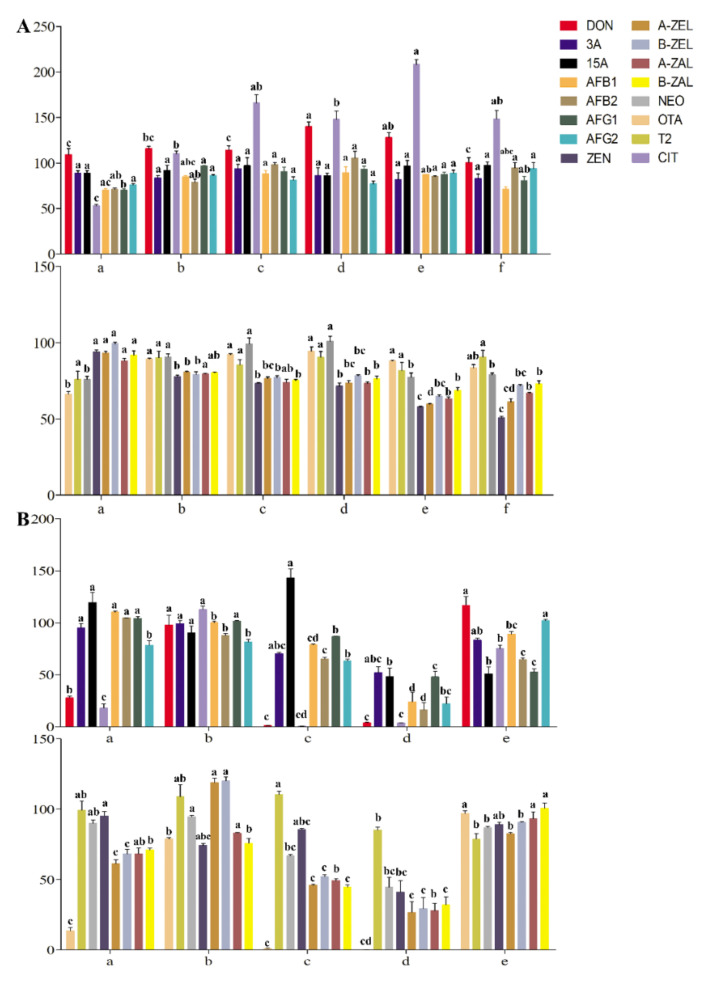
Optimization of extraction and purification methods: (**A**) The recoveries of mycotoxins in tea matrix by acetonitrile with different formic acid content; (a) acetonitrile; (b) acetonitrile with 1% formic acid; (c) acetonitrile with 2% formic acid; (d) acetonitrile with 5% formic acid; (e) acetonitrile with 10% formic acid; (f) acetonitrile with 15% formic acid. (**B**) The Recoveries of mycotoxins in tea matrix by different purification methods; (a) Cleanert MC; (b) MWCNTs-COOH:HLB:SG (1:7.5:7.5); (c) CNW 301 MFC; (d) MFC260; (e) Romerlab MycoSpin 400. Different low case letters above columns indicate statistical differences at *p* < 0.05; The bars (*n* = 3) represent mean ± SE.

**Figure 3 toxins-14-00169-f003:**
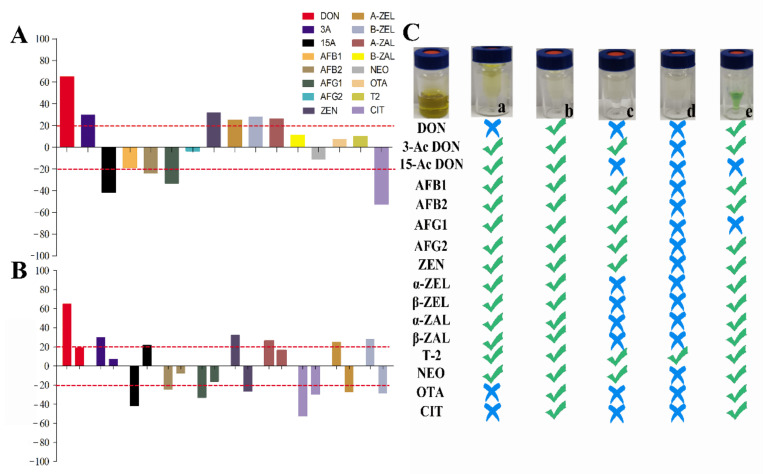
Matrix effects and purification effects of 16 mycotoxins after extraction and purification: (**A**) Matrix effects of 16 mycotoxins extracted by acetonitrile with 1% formic acid. (**B**) Matrix effects of 10 mycotoxins (matrix effects are unacceptable after extraction) before and after purification. (**C**) Purification effects for 16 mycotoxins with different clean-up columns; (a) Cleanert MC; (b) MWCNTs-COOH:HLB:SG (1:7.5:7.5); (c) CNW 301 MFC; (d) MFC260; (e) Romerlab MycoSpin 400; symbols “√” represent recoveries ranging from 60% to 120%, while symbols “×” represent recoveries <60% or >120%, respectively.

**Table 1 toxins-14-00169-t001:** Comparison of multi-mycotoxins in tea determination by HPLC or LC-MS/MS recently reported.

Detection (Mycotoxins)	Extraction//Purification (The Chromatographic Column)	Matrix (Recoveries)	LODs//LOQs µg·kg^−1^
UPLC-MS/MS (22 Mycotoxins) [19]	1% formic acid-EtOAc//NH2-SPE, C18-SPE (Acquity UPLC BEH C18)	Raw tea materials (China and Belgium) (91–107%)	2.1–122// 4.1–243
HPLC-MS/MS-IT (16 Mycotoxins) [18]	DLLME: NaCl, EtOAc, ACN, MeOH, CHCL3 (Gemini-NX column C18)	Black, red, green tea beverages (Spain) (65–127%)	0.05–10.0// 0.2–33.0
HPLC-FD (5 Mycotoxins) [17]	80% MeOH, NaCl//AFs-IAC, OTA-IAC (ZORBAX Eclipse XDB C18)	Black, green tea (Iran) (74.1–99.6%)	0.1–0.47// 0.4–1.23
HPLC (10 Mycotoxins) [14]	ACN or MeOH//water containing NaCl//MFC-IAC (C18 column Xbridge)	Dark tea (China) (76.8–95.6%)	0.018–34.4// Not Found
UPLC-MS/MS (7 Mycotoxins) [22]	1.0 mol/L ammonium acetate, 98% ACN-DMSO// MgSO4, C18 (Acquity HSS-T3 column)	Black tea (China) (75.20–124.4%)	Not Found// 5
UPLC-MS/MS (16 Mycotoxins) This work	1% formic acid -ACN//MWCNTs-COOH, HLB, SG (Agilent Extend C18)	Green, oolong, black, and dark tea (China) (61.27–118.46%)	0.015–15.00// 0.03–30.00

**Table 2 toxins-14-00169-t002:** Retention time and MS parameters for the analysis of mycotoxins.

Analytes	Molecular Weight	TR (min)	Molecular Ion	ESI	Parent Ions (*m*/*z*)	Product Ions (*m*/*z*)	CE (eV)
AFB_1_	312.27	10.82	[M + H]^+^	ESI^+^	313.110	241.100 ^a^	44
					313.120	285.200	23
					313.130	213.100	46
AFB_2_	314.29	10.55	[M + H]^+^	ESI^+^	315.110	287.200 ^a^	30
					315.120	259.300	33
					315.130	243.200	40
AFG_1_	328.27	10.18	[M + H]^+^	ESI^+^	329.110	199.000	58
					329.120	243.300	32
					329.130	200.100 ^a^	41
AFG_2_	330.29	9.86	[M + H]^+^	ESI^+^	331.110	189.100 ^a^	42
					331.120	245.100	34
					331.130	314.200	25
ZEN	318.36	13.17	[M − H]^−^	ESI^−^	317.110	175.030 ^a^	25
					317.120	131.020	31
					317.130	273.170	20
α-ZEL	320.38	12.90	[M − H]^−^	ESI^−^	319.110	275.200 ^a^	21
					319.120	160.000	33
					319.130	301.200	23
β-ZEL	320.38	12.36	[M − H]^−^	ESI^−^	319.110	275.200 ^a^	21
					319.120	160.000	33
					319.130	301.200	23
α-ZAL	322.40	12.74	[M − H]^−^	ESI^−^	321.110	277.200 ^a^	23
					321.120	303.200	22
					321.130	259.200	25
β-ZAL	322.40	12.10	[M − H]^−^	ESI^−^	321.110	277.200 ^a^	23
					321.120	303.200	22
					321.130	259.200	25
DON	296.32	6.55	[M + CH_3_COO]^−^	ESI^−^	355.000	265.000 ^a^	17
					355.100	247.200	22
15-Ac DON	338.35	9.41	[M + NH_4_]^+^	ESI^+^	356.000	137.000 ^a^	5
					356.100	321.000	13
3-Ac DON	338.35	9.46	[M + CH_3_COO]^−^	ESI^−^	397.000	307.160 ^a^	16
					397.100	173.100	15
OTA	403.81	11.96	[M + H]^+^	ESI^+^	404.110	105.100	18
					404.120	221.000	35
					404.130	239.100 ^a^	25
NEO	382.40	8.09	[M + NH_4_]^+^	ESI^+^	400.100	185.100 ^a^	16
					400.120	215.100	14
T-2	466.52	12.51	[M + NH_4_]^+^	ESI^+^	484.110	215.000 ^a^	20
					484.120	165.000	66
					484.130	197.000	24
CIT	250.25	11.28	[M + OH]^−^	ESI^−^	267.110	221.000	20
					267.120	177.000	28
					267.130	249.000 ^a^	21

RT: retention time; CE: collision energy; ^a^ Quantifying ions.

**Table 3 toxins-14-00169-t003:** Overview of the accuracy and precision of the developed LC-MS/MS method.

Targets	Spiked Recovery (%) RSD (%, *n* = 6)												Precision (RSD, %)	
	Green Tea			Oolong Tea			Black Tea			Dark Tea			RSDr	RSDR
	Low	Middle	High	Low	Middle	High	Low	Middle	High	Low	Middle	High	(*n* = 6)	(*n* = 6)
AFB1	92.90	101.72	65.51	86.60	95.47	68.48	90.99	94.08	63.27	92.65	99.67	74.88	94.26	108.08
	13.75	7.81	4.55	7.76	5.84	3.49	5.97	2.67	3.17	8.06	10.45	4.00	4.75	3.36
AFB2	90.28	100.87	62.20	79.70	86.54	65.41	84.60	86.57	66.74	80.99	98.44	72.24	88.50	102.82
	17.77	11.12	3.55	10.01	5.68	4.61	3.50	6.60	3.12	15.66	10.27	3.09	8.11	6.46
AFG1	107.01	109.72	64.29	115.83	105.93	69.55	104.52	104.10	65.24	114.44	101.74	77.21	107.91	109.54
	10.72	8.55	4.12	4.17	2.58	4.75	8.46	6.92	4.37	6.04	4.38	5.69	4.36	4.79
AFG2	89.13	88.61	63.93	77.31	99.96	102.65	64.48	114.39	87.49	77.02	98.27	93.81	88.65	101.26
	19.39	8.76	7.89	19.34	5.87	11.72	1.19	19.13	17.04	19.95	15.80	18.75	16.59	4.69
ZEN	76.58	91.47	63.55	78.48	88.46	71.93	71.46	76.65	61.81	69.62	73.45	60.79	78.27	85.55
	8.19	10.06	4.40	8.57	10.16	7.57	5.96	3.50	2.68	8.35	3.53	0.70	9.35	8.99
α-ZEL	117.62	99.86	85.93	116.21	98.63	95.79	100.79	89.30	78.86	110.90	97.93	86.53	103.91	88.12
	1.68	6.21	18.97	2.95	7.31	18.74	14.34	4.46	13.34	8.72	10.77	16.89	8.95	9.67
β-ZEL	88.37	69.75	94.55	101.14	67.84	103.51	77.20	62.80	91.34	85.42	64.25	101.90	77.10	78.42
	7.34	11.08	17.71	11.50	8.84	12.59	3.94	3.18	11.48	7.88	6.63	14.62	16.43	8.29
α-ZAL	74.79	79.36	86.95	72.93	74.65	100.73	76.86	71.90	92.19	73.71	71.40	100.65	74.45	73.12
	9.12	6.52	18.66	6.52	9.06	16.70	5.86	4.25	19.28	7.44	3.35	16.01	3.32	8.62
β-ZAL	67.43	69.41	89.46	62.71	67.92	101.03	64.14	61.27	86.52	63.42	63.26	97.79	64.95	70.39
	6.79	9.61	19.17	4.03	8.25	14.71	9.31	1.56	18.90	9.36	5.08	15.91	4.19	12.55
DON	75.45	90.96	108.16	103.11	98.11	112.90	83.49	98.84	104.18	66.74	90.48	106.39	88.40	97.25
	13.81	4.13	6.56	13.36	7.65	5.65	7.51	4.61	7.96	11.90	4.53	9.15	13.21	15.22
15AcDON	101.34	101.17	76.59	87.24	82.43	83.40	84.65	77.16	74.57	95.42	100.72	87.43	91.27	95.57
	10.58	19.74	12.89	19.73	18.16	9.68	14.51	18.03	11.06	19.48	18.17	10.59	9.82	9.04
3AcDON	101.45	99.22	96.07	105.69	101.47	100.25	100.10	108.85	95.55	97.13	102.95	90.57	102.11	107.59
	7.09	3.80	12.89	7.21	3.42	7.77	5.70	3.59	6.16	8.08	4.84	15.04	3.41	3.48
OTA	90.02	90.26	77.02	80.60	66.11	92.40	64.36	64.80	90.64	68.73	64.18	97.53	73.63	97.17
	18.90	19.72	11.19	12.99	6.46	8.15	4.46	0.94	8.32	11.26	4.13	11.77	14.63	8.34
NEO	75.32	99.07	96.55	100.04	106.40	109.62	74.58	76.63	102.32	67.99	77.98	100.31	84.75	100.54
	14.53	17.79	15.07	14.05	12.87	8.88	14.59	8.69	9.43	11.52	7.76	9.87	16.12	11.06
T-2	66.54	104.40	88.89	81.20	93.91	100.05	80.21	88.71	102.24	65.51	92.03	104.72	84.06	96.23
	11.08	13.56	16.89	17.63	9.14	14.55	9.66	4.20	4.15	10.08	6.01	4.58	14.99	13.28
CIT	103.01	105.01	61.89	114.85	111.90	77.47	118.46	111.03	65.41	101.74	74.15	61.50	105.02	97.45
	7.66	10.15	4.05	4.86	5.94	10.03	2.29	3.44	7.94	11.81	15.04	2.23	12.27	9.68

Low, Middle, and High represent the spiked low, middle, and high concentrations of mycotoxins respectively. RSD: repeatability for recoveries in each type of tea samples at each fortified concentration; RSDr: intraday precision (repeatability) in mycotoxin-fortified tea samples; RSDR: interday precision (reproducibility) in mycotoxin-fortified tea samples.

**Table 4 toxins-14-00169-t004:** Concentration range and mean of mycotoxins in all tea samples (µg·kg^−1^).

Mycotoxin	Green Tea			Oolong Tea			Black Tea			Dark Tea			TSP	rMRL_S_
	Min	Max	Mean	Min	Max	Mean	Min	Max	Mean	Min	Max	Mean		
AFB1	<LOD	<LOD	ND	<LOD	<LOD	ND	<LOD	<LOD	ND	0.97	1.11	1.04	(0,0,0,0)/80	5 ^a^
AFB2	<LOD	<LOD	ND	0.16	0.61	0.39	<LOD	0.15	0.15	0.13	0.32	0.22	(0,0,0,0)/80	5 ^b^
AFG1	0.06	0.07	0.07	0.19	0.91	0.47	0.43	0.87	0.62	0.38	0.84	0.61	(0,0,0,0)/80	5 ^b^
AFG2	0.90	6.75	2.97	2.98	13.36	8.71	0.79	7.51	3.46	3.27	23.49	8.98	(1,15,6,15)/80	5 ^b^
AFs	0.90	6.75	2.98	2.98	14.27	9.03	1.22	7.51	3.71	0.20	24.07	8.83	(0,0,0,1)/80	20 ^a^
ZEN	<LOD	0.26	0.26	<LOD	<LOD	ND	<LOD	<LOD	ND	<LOD	1.44	1.44	(0,0,0,0)/80	75 ^c^
α-ZEL	10.30	24.71	19.72	29.27	48.03	39.77	19.19	116.22	51.28	25.31	71.09	36.55	(0.0,5,0)/80	75 ^d^
β-ZEL	20.49	34.95	25.98	40.94	96.08	57.54	26.62	226.24	90.21	25.21	62.78	45.38	(0,2,6,0)/80	75 ^d^
α-ZAL	<LOD	<LOD	ND	<LOD	12.87	12.87	<LOD	<LOD	ND	<LOD	<LOD	ND	(0,0,0,0)/80	75 ^d^
β-ZAL	<LOD	<LOD	ND	<LOD	<LOD	ND	<LOD	<LOD	ND	<LOD	<LOD	ND	(0,0,0,0)/80	75 ^d^
DON	11.42	89.89	42.72	667.35	1181.48	958.58	247.40	508.99	423.59	176.02	328.94	253.56	(0,5,0,0)/80	750 ^c^
15AcDON	<LOD	<LOD	ND	<LOD	<LOD	ND	77.40	707.57	294.25	<LOD	425.70	192.65	(0,0,0,0)/80	750 ^e^
3Ac DON	<LOD	<LOD	ND	<LOD	8.03	6.71	<LOD	<LOD	ND	<LOD	<LOD	ND	(0,0,0,0)/80	750 ^e^
OTA	<LOD	<LOD	ND	<LOD	1.07	0.34	0.53	2.81	1.52	0.30	11354.64	666.45	(0,0,0,4)/80	5 ^c^
NEO	<LOD	<LOD	ND	<LOD	1.30	0.91	1.28	4.61	2.50	3.49	13.56	7.79	(0,0,0,0)/80	200 ^f^
T-2	<LOD	<LOD	ND	<LOD	<LOD	ND	<LOD	<LOD	ND	<LOD	<LOD	ND	(0,0,0,0)/80	200 ^c^
CIT	4.16	9.41	6.73	4.84	91.90	25.91	8.90	93.27	54.59	12.20	203.76	62.76	(0,0,0,1)/80	200 ^g^

ND: not detectable; LOD: limit of detection; TSP: Number of positive samples in green tea, oolong tea, black tea, or dark tea samples (n_i_)/total samples (N). r-MRLs: relevant-Maximum Regulation Limits. ^a^ Limits for AFB1 and AFs in materials used for herbal tea infusions set by Argentina; ^b^ MRLs with respect to AFB1 MRL: 5 µg·kg^−1^; ^c^ MRLs of contaminants of foods in EU (EC Regulation No 1881/2006); ^d^ MRLs with respect to ZEN MRL: 75 µg·kg^−1^; ^e^ MRLs with respect to DON MRL: 750 µg·kg^−1^; ^f^ MRLs with respect to T-2 MRL: 200 µg·kg^−1^. ^g^ MRLs with respect to the evaluation of citrinin occurrence in Chinese Liupao tea by Li et al. [34].

## Data Availability

The data that support the findings of this study are available from the corresponding author upon reasonable request.

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
