# Peer review of "Development of a Novel UPLC-MS/MS Method for the Simultaneous Determination of 16 Mycotoxins in Different Tea Categories"

_toxins, 2022, doi:10.3390/toxins14030169_

Round 1
Reviewer 1 Report
The manuscript describes an UPLC-MS/MS method using a home-made purification to detect 16 mycotoxins simultaneously in different tea samples.
The manuscript was well-structured, with clear objectives and a methodology that allows the authors to achieve the goals. However, there are some critical points that should be addressed. For instance, the statistical analysis is poorly discussed throughout the paper. In addition, the details of statistical analysis should be included in the figures. Also, the authors could consider grammar and spell-check the text in order to avoid some minor mistakes.
Major comments:
Title:
- I suggest a more succinct title such as “Development of a novel UPLC-MS/MS method for the simultaneous determination of 16 mycotoxins in different tea categories”.
Abstract:
-In line 28, the abstract could be improved by adding a concluding paragraph.
Introduction:
-Line 64; the objective must be more clear and concise. I suggest: The present study aimed to develop and validate an ultra-performance LC-MS/MS method with a home-made purification column to determine 16 mycotoxins in four categories of tea (green, oolong, black, and dark tea).
Results and discussion:
-In line 76, the authors could amend the sentence to…С18 reversed-phase liquid chromatography columns are the most commonly used columns for the separation of mycotoxins
- Why should I use this methodology and not use another already published methodology? What is the difference between the methods? There is no doubt that these questions should be addressed in the discussion section. Furthermore, the answers should be based on an appropriate statistical analysis.
Material and methods:
-In line 279: The number of samples in each stage of repiling could be cited—for instance, the first stage (n= 10).
-In line 287, remove (i.e.).
-In line 288, change 4000 r min-1 to G-force throughout the manuscript.
-In line 292, consider change “Ultimately” to Finally.”
-Consider modifying the sentence in line 294 “ Compare and evaluate… This sentence is not clear.
-Line 298: please cite the literature or the industrial product used to purify the samples to improve the work's reproducibility.
-The authors cited "intermediate concentrations" twice in the manuscript, and it is unclear what concentrations the authors are referring to in relation to mycotoxins. Is it the same concentration as expressed in Table s1? If so, perhaps the authors should consider referencing it.
-Please separate °C from the number.
-Line 327, the sentence: “The range of added standard concentrations to mycotoxin-free tea 327 with six replicates was 0.13~1200 μg·kg-1 (Table S1 and Figure S1)” is unclear… The authors could explain whether the added concentrations had six replicates in each point ranging from 0.13 to 1200 ug kg-1? Or, six different points were performed ranging from 0.13 to 1200 μg·kg-1? If the second case is correct, what is the number of replicates for each point on the curve?
-section 4.7 data analysis must be improved: the statistical analysis carried out to process the data should be cited.
Conclusion:
- The conclusions need to be improved. The authors cannot conclude that the process samples of ripened Pu-erh were safe, as the risk of mycotoxin concentrations to human health was not assessed. I suggest adding a paragraph comparing the different purification columns as well as the methodology employed.
Figures:
-Figure 1: the graph does not provide any relevant information regarding the study; the authors could consider modifying the image by keeping the Data Analysis image (chromatogram) and changing the figure caption.
-Figure 2: The statistical parameters should be included. Do the bars represent mean +/- SD or SE? Is there is a difference among formic acid concentrations? How can we choose the better method? n=?
-Figure 3: The statistical parameters should be included. For example, the bar of errors, number of replicates and statistical analysis. Do the values represent averages?
Author Response
Dear Reviewer,
Thanks for your comments and suggestions to our manuscript (toxins-1589311). All the issues raised have been carefully addressed and all the changes have been clearly marked in the new revision. Here below are our descriptions for revising according to the point-to-point comments.
Yours sincerely,
Na Liu
Comments and Suggestions for Authors:
The manuscript describes an UPLC-MS/MS method using a home-made purification to detect 16 mycotoxins simultaneously in different tea samples. The manuscript was well-structured, with clear objectives and a methodology that allows the authors to achieve the goals. However, there are some critical points that should be addressed. For instance, the statistical analysis is poorly discussed throughout the paper. In addition, the details of statistical analysis should be included in the figures. Also, the authors could consider grammar and spell-check the text in order to avoid some minor mistakes.
Response to Reviewer:
Thank you for your suggestions, the problems with details of statistical analysis in figures and discussion have been thoroughly checked and revised throughout the manuscript (Lines 138-152). We have improved the full text from beginning to the end and marked changes using the “Track Changes” function. Here below are our specific descriptions according to the point-to-point comments:
Major comments:
Comment 1-Title:
-I suggest a more succinct title such as “Development of a novel UPLC-MS/MS method for the simultaneous determination of 16 mycotoxins in different tea categories”.
Response to Reviewer comment on Title (NO. 1):
-Thanks for your suggestion, we have revised the title as you suggested in the new revision (Lines 2-5).
Comment 2-Abstract:
-In line 28, the abstract could be improved by adding a concluding paragraph.
Response to Reviewer comment on Abstract (NO. 2):
-We have added a concluding paragraph into the section of abstract. You can refer to lines 25-26 in the new revision.
Comment 3-Introduction:
-Line 64; the objective must be more clear and concise. I suggest: The present study aimed to develop and validate an ultra-performance LC-MS/MS method with a home-made purification column to determine 16 mycotoxins in four categories of tea (green, oolong, black, and dark tea).
Response to Reviewer comment on Introduction (NO. 3):
-We have revised the objective in the section of introduction as you suggested. You can refer to lines 75-81 in the revised manuscript.
Comment 4-Results and discussion:
-In line 76, the authors could amend the sentence to…С18 reversed-phase liquid chromatography columns are the most commonly used columns for the separation of mycotoxins
Response to Reviewer comment on Results and discussion (NO. 4):
-According to your suggestion, we have revised the sentence as “C18 reversed-phase liquid chromatography columns are the most commonly used columns for the separation of mycotoxins.” (Lines 87-89)
Comment 5-Results and discussion:
-Why should I use this methodology and not use another already published methodology? What is the difference between the methods? There is no doubt that these questions should be addressed in the discussion section. Furthermore, the answers should be based on an appropriate statistical analysis.
Response to Reviewer comment on Results and discussion (NO. 5):
-We have carefully revised and improved the results and discussion section, including the detailed comparison between this developed method and other already published methodologies based on the obtained results from statistical analysis. You can refer to the changes in the lines 141-142, 174-178, and lines 200-202.
Comment 6-Material and methods:
-In line 279: The number of samples in each stage of repiling could be cited—for instance, the first stage (n= 10).
Response to Reviewer comment on Material and methods (NO. 6):
-We have illuminated the number of samples in each stage of repiling. You can refer to the changes in the revised manuscript (Lines 322-324) and Table S2.
Comment 7-Material and methods:
-In line 287, remove (i.e.).
Response to Reviewer comment on Material and methods (NO. 7):
-We have removed “i.e.” in the parentheses (Line 329).
Comment 8-Material and methods:
-In line 288, change 4000 r min-1 to G-force throughout the manuscript.
Response to Reviewer comment on Material and methods (NO. 8):
-We have changed the rpm to RCF throughout the manuscript. You can refer to the lines 330-331 and lines 334-335 in the new revision.
Comment 9-Material and methods:
-In line 292, consider change “Ultimately” to Finally.”
Response to Reviewer comment on Material and methods (NO. 9):
-We have changed “Ultimately” to Finally.” Please refer to line 305 in the revised manuscript (line 335).
Comment 10-Material and methods:
-Consider modifying the sentence in line 294 “Compare and evaluate… This sentence is not clear.
Response to Reviewer comment on Material and methods (NO. 10):
-We have modified the sentence for clarity, “The recoveries of mycotoxins purification and extraction from tea matrices were compared and evaluated.”. You can refer to line 337-339 in the new revision.
Comment 11-Material and methods:
-Line 298: please cite the literature or the industrial product used to purify the samples to improve the work's reproducibility.
Response to Reviewer comment on Material and methods (NO. 11):
-We have added the reference of literature and the operation instruction to improve the work's reproducibility. You can refer to the lines 342-343.
Comment 12-Material and methods:
-The authors cited "intermediate concentrations" twice in the manuscript, and it is unclear what concentrations the authors are referring to in relation to mycotoxins. Is it the same concentration as expressed in Table s1? If so, perhaps the authors should consider referencing it.
Response to Reviewer comment on Material and methods (NO. 12):
-Thanks for your comment. The "intermediate concentrations" in the manuscript are the same concentrations as “middle concentrations” in Table S1. We have revised "intermediate concentrations" as “middle concentrations” and referenced the Table S1 where "middle concentrations" first appeared. You can refer to the line 104 and line 353 in the new revision.
Comment 13-Material and methods:
-Please separate °C from the number.
Response to Reviewer comment on Material and methods (NO. 13):
-We have revised throughout the manuscript. Please refer to lines 313, 340, 350, and 359.
Comment 14-Material and methods:
-Line 327, the sentence: “The range of added standard concentrations to mycotoxin-free tea with six replicates was 0.13~1200 μg·kg-1 (Table S1 and Figure S1)” is unclear… The authors could explain whether the added concentrations had six replicates in each point ranging from 0.13 to 1200 ug kg-1? Or, six different points were performed ranging from 0.13 to 1200 μg·kg-1? If the second case is correct, what is the number of replicates for each point on the curve?
Response to Reviewer comment on Material and methods (NO. 14):
-The added concentrations had six replicates in each point ranging from 0.13 to 1200 μg·kg-1 for 16 mycotoxins at low, middle, and high concentration levels (Table S1). We have moved the sentence to line 343 “The range of added standard concentrations to mycotoxin-free tea with six replicates was 0.13~1200 µg·kg-1 (Table S1 and Figure S1).” to make it easier to understand the description of intra- or inter-day precision.
Comment 15-Material and methods:
-section 4.7 data analysis must be improved: the statistical analysis carried out to process the data should be cited.
Response to Reviewer comment on Material and methods (NO. 15):
-We have improved the section 4.7 “Data analysis”. You can refer to lines 392-395 in the revised manuscript.
Comment 16-Conclusion:
-The conclusions need to be improved. The authors cannot conclude that the process samples of ripened Pu-erh were safe, as the risk of mycotoxin concentrations to human health was not assessed. I suggest adding a paragraph comparing the different purification columns as well as the methodology employed.
Response to Reviewer comment on Conclusion: (NO. 16):
-According to your suggestion, we have improved the conclusions. Please refer to the lines 270-273, 275-276, and 278-281.
Comment 17-Figures:
-Figure 1: the graph does not provide any relevant information regarding the study; the authors could consider modifying the image by keeping the Data Analysis image (chromatogram) and changing the figure caption.
Response to Reviewer comment on Figures (NO. 17):
-We have modified the image by keeping the Data Analysis image (chromatogram) as you suggested and changed the figure caption as “Total ion chromatograms of 16 mycotoxins at middle concentration under optimized chromatographic and mass spectrometry conditions” (Lines 118-122).
Comment 18-Figures:
-Figure 2: The statistical parameters should be included. Do the bars represent mean +/- SD or SE? Is there is a difference among formic acid concentrations? How can we choose the better method? n=?
Response to Reviewer comment on Figures (NO. 18):
-We have added the statistical parameters in the Figure 2 (Lines 143-145). Meanwhile, the related descriptions were supplemented below in the Figure 2 (Lines 151-152), and the new revision (Lines 138-140). We choose the better approach by considering two aspects: The mycotoxin recoveries were calculated and evaluated firstly according to the document and commission decision (SANTE/12682/2019, 2002/657/EC). Recoveries <60% or >120% are not chosen, while recoveries ranging from 60% to 120% represent acceptable. Additionally, mycotoxin recoveries for each solvent mixture and purification method were evaluated using two-way ANOVA at a significance level of 0.05; and Tukey's multiple comparisons testing was used to evaluate the significance of difference.
Comment 19-Figures:
-Figure 3: The statistical parameters should be included. For example, the bar of errors, number of replicates and statistical analysis. Do the values represent averages?
Response to Reviewer comment on Figures (NO. 19):
The columns represent a ratio of two slopes from standard curve, which is consistent with the SANTE/12682/2019 document and other previous studies [1-3]. As described in this manuscript, “the ME was evaluated by analyzing the slopes of two sets of mycotoxin standards prepared with organic solvent and the relevant tea extracts. MEs were calculated according to Eq. 1 below, where Ka (matrices) and Kb (organic solvent) are the slopes of curves, respectively.” (Lines 363-366),
[1] Dong, H., Xian, Y., Xiao, K., Wu, Y., Zhu, L., & He, J. (2019). Development and comparison of single-step solid phase extraction and QuEChERS clean-up for the analysis of 7 mycotoxins in fruits and vegetables during storage by UHPLC- MS/MS. Food Chem, 274, 471-479. https://doi.org/10.1016/j.foodchem.2018.0 9.035.
[2] Kun, Wang, Kunde, Li, Xinwen, & Huang, et al. (2017). A simple and fast extraction method for the determination of multiclass antibiotics in eggs using LC-MS/MS. Journal of Agricultural and Food Chemistry, 65(24), 5064-5073.
[3] SANTE 12682/2019. Guidance document on method validation and quality control procedures for pesticide residues analysis in food and feed. 2019.

Reviewer 2 Report
The Authors conducted interesting research related to mycotoxins content in different tea categories. The manuscript is well written. However, there are some points that have to be corrected. Comments are listed below:
There are 7 groups of mycotoxins in the title, however, there are not precisely indicated in the text or Tables. For example in Table 2, the Authors should add a column next to “Analytes” with the group of mycotoxins and classify listed mycotoxins to the right group.
L27: Camellia sinensis write in italics
In Table 2 and its footnote: retention time is mostly determined as RT, not TR
Figure 1 caption: … description of mycotoxins determination …
Figure 2: Statistical significance between mycotoxin recovery for each solvent mixture is needed. And between mycotoxin recovery for each purification method. Because some of data seem to be not significant.
L130: Write heading in italics
L137: “Our home-made purification method” write in the brackets at the end of this statement MWCNTs-COOH:HLB:SG
L138-149: It is not necessary to repeat “Our home-made purification method”. Instead that, write MWCNTs-COOH:HLB:SG
L191: C. sinensis write in italics
L239-241: Some MRLs were exceeded (for DON, OTA, α - ZEL, β – ZEL, AFG2, AFs)
L241-243: The Authors should include that level of mycotoxins is influenced by fermentation. Green tea, which is not fermented has lower mycotoxins content, compared to other teas.
L285: A total of 1.0 g sample was placed …
L291: Write ratio of column components
L310: What do you mean by “different argon gases”?
In Figure S1 caption write “chromatogram” instead “chromatography”
Author Response
Dear Reviewer,
Thanks for your comments and suggestions to our manuscript (toxins-1589311). All the issues raised have been carefully addressed and all the changes have been clearly marked in the new revision. Here below are our descriptions for revising according to the point-to-point comments.
Yours sincerely,
Na Liu
Comments and Suggestions for Authors:
The Authors conducted interesting research related to mycotoxins content in different tea categories. The manuscript is well written. However, there are some points that have to be corrected. Comments are listed below:
Response to Reviewer:
Comment 1:
There are 7 groups of mycotoxins in the title, however, there are not precisely indicated in the text or Tables. For example, in Table 2, the Authors should add a column next to “Analytes” with the group of mycotoxins and classify listed mycotoxins to the right group.
Response 1:
We have precisely indicated the classification for 7 groups of mycotoxins in the text. Please refer to lines 31-33 in the revised manuscript.
Comment 2:
L27: Camellia sinensis write in italics
Response 2:
We have written “Camellia sinensis” in italics. You can refer to line 38 in the new revision.
Comment 3:
In Table 2 and its footnote: retention time is mostly determined as RT, not TR
Response 3:
We have corrected the footnote in the Table 2. You can refer to the line 116 in the revised manuscript.
Comment 4:
Figure 1 caption: … description of mycotoxins determination …
Response 4:
We have modified the image by keeping the Data Analysis image (chromatogram) for understanding and changed the figure caption as “Total ion chromatograms of 16 mycotoxins at middle concentration under optimized chromatographic and mass spectrometry conditions” (Lines 120-122).
Comment 5:
Figure 2: Statistical significance between mycotoxin recovery for each solvent mixture is needed. And between mycotoxin recovery for each purification method. Because some of data seem to be not significant.
Response 5:
We have added the statistical parameters and the related descriptions in the Figure 2. Mycotoxin recoveries for each solvent mixture and purification method were evaluated using two-way ANOVA at a significance level of 0.05; and Tukey's multiple comparisons testing was used to evaluate the significance of difference. You can refer to the Figure 2 or Lines 143-145, and lines 151-152.
Comment 6:
L130: Write heading in italics
Response 6:
We have carefully checked and revised in accordance with the template of this journal (Lines 124, and 154).
Comment 7:
L137: “Our home-made purification method” write in the brackets at the end of this statement MWCNTs-COOH: HLB: SG.
Response 7:
We have written “Our home-made purification method” in the brackets at the end of “MWCNTs-COOH: HLB: SG”. You can refer to the lines 161-162 in the new revision.
Comment 8:
L138-149: It is not necessary to repeat “Our home-made purification method”. Instead that, write MWCNTs-COOH: HLB: SG.
Response 8:
“MWCNTs-COOH: HLB: SG” have been written instead of “Our home-made purification method” as you suggested. Please refer to the line 163 in the revised manuscript.
Comment 9:
L191: C. sinensis write in italics
Response 9:
We have written “C. sinensis” in italics. You can refer line 212
Comment 10:
L239-241: Some MRLs were exceeded (for DON, OTA, α - ZEL, β – ZEL, AFG2, AFs)
Response 10:
We have modified the sentence as “Some MRLs were exceeded (for DON, OTA, α-ZEL, β-ZEL, AFG2, or AFs) by the detected contents in some samples” (Lines 278-279).
Comment 11:
L241-243: The Authors should include that level of mycotoxins is influenced by fermentation. Green tea, which is not fermented has lower mycotoxins content, compared to other teas.
Response 11:
According to your suggestion, we have expanded the cause of mycotoxin contamination in terms of tea types, including the differences on fermented processes. You can refer to the line 283 in the revised manuscript.
Comment 12:
L285: A total of 1.0 g sample was placed …
Response 12:
We have revised the sentence in the manuscript as you suggested (Line 327).
Comment 13:
L291: Write ratio of column components
Response 13:
We have changed as “MWCNTs-COOH: HLB: SG”. You can refer to lines 333-334 in the new revision.
Comment 14:
L310: What do you mean by “different argon gases”?
Response 14:
We have revised the sentence as “…different collision energies with argon…”. You can refer to the line 355 in the new revision.
Comment 15:
In Figure S1 caption write “chromatogram” instead “chromatography”
Response 15:
We have revised as “chromatogram”. Please refer to the caption of Figure S1.

Round 2
Reviewer 1 Report
The authors have revised the manuscript and improved its quality by addressing some critical points such as statistical analysis. Reading the text I noticed that the authors addressed an economic point, arguing that their product is more affordable than commercial ones. Therefore, I think that to clarify this issue, the authors could add some information about the price of their product compared to the commercial one in the discussion section.
Minor comments:
- Table s1: Include number of sample information.
- Table s2; Include information about concentration of mycotoxins.
Author Response
Dear Reviewer,
Thanks for your comments and suggestions to our manuscript (toxins-1589311). All the issues raised have been carefully addressed and all the changes have been clearly marked in the new revision. Here below are our descriptions for revising according to the point-to-point comments.
Yours sincerely,
Na Liu
Comments and Suggestions for Authors:
The authors have revised the manuscript and improved its quality by addressing some critical points such as statistical analysis. Reading the text, I noticed that the authors addressed an economic point, arguing that their product is more affordable than commercial ones. Therefore, I think that to clarify this issue, the authors could add some information about the price of their product compared to the commercial one in the discussion section.
Response to Reviewer comments:
Thank you for your suggestions, we carefully checked the cost price of the Cleanert MC (22.00 CNY), MFC 301 (84.00 CNY), MFC 260 (72.00 CNY), MycoSpin 400 (103.52 CNY), and the MWCNTs-COOH: HLB: SG (6.09 CNY) for purifying one tea sample respectively. Therefore, our home-made column is the lowest cost. We have added the information about the cost price of our home-made column compared to other commercial purification columns in the discussion section of the new revision (Lines 178-181). We have carefully improved supplementary materials and marked changes with highlight in green. Here below are our specific descriptions according to the point-to-point comments:
Minor comments:
Comment 1-Table S1:
Table s1: Include number of sample information.
Response to Reviewer comment on Table S1 (NO. 1):
-Thanks for your suggestion, we have included the number of sample information for table S1. Please refer to the changes in the new revised supplementary materials.
Comment 2-Table S2:
Table s2; Include information about concentration of mycotoxins.
Response to Reviewer comment on Table S2 (NO. 2):
-We have added information on mycotoxin concentration units. You can refer to table S2 in the new revised supplementary materials.
